# Addressing Health Illiteracy and Stunting in Culture-Shocked Indigenous Populations: A Case Study of Outer Baduy in Indonesia

**DOI:** 10.3390/ijerph21091114

**Published:** 2024-08-23

**Authors:** Liza Diniarizky Putri, Herlina Agustin, Iriana Bakti, Jenny Ratna Suminar

**Affiliations:** Faculty of Communication Sciences, Universitas Padjadjaran, Bandung 45363, Indonesia; h.agustin@unpad.ac.id (H.A.); iriana.bakti@unpad.ac.id (I.B.); jenny.suminar@unpad.ac.id (J.R.S.)

**Keywords:** stunting, health literacy, health communication, Indigenous communities, Baduy tribe

## Abstract

This research aims to determine the factors, impacts, and solutions for health literacy in the Outer Baduy hamlets of Kanekes Village, Indonesia. The method used in this research is qualitative, which produces an in-depth explanation of the existing problems. Data were collected through interviews and documentation. Interviews were conducted with key figures, including two female Baduy residents with stunted children, one retainer, the head of the NGO SRI, a midwife who works in the Baduy village, and the head of the Lebak social service. Apart from that, secondary data in the form of recordings of community service talk shows conducted by the University of Indonesia to overcome stunting in Baduy were also analysed. The results show that the factors associated with the low health literacy of the Baduy community are literacy, writing and reading, taboos on eating certain foods, people spending too much time in the fields, people learning by imitating their parents, demanding access to villages, lack of consistency from external parties in providing health programs, and gender segregation in Baduy society. The impact of the low health literacy of the Baduy community is fatalism, high maternal and child mortality rates, and high health costs. The proposed strategies for increasing the health literacy of the Baduy community based on the findings of this research include developing health literacy by targeting community leaders, managing information-technology-based health-information groups, and always presenting at least one health worker among the residents who provides an example of healthy living, encouraging collective reflection. when health cases occur, and balancing gender communication.

## 1. Introduction

Many countries struggle with high levels of health illiteracy and low effectiveness of health campaigns for their citizens [1]. This issue is particularly pronounced among Indigenous populations, who are often targeted by educational activities designed to reduce health illiteracy [2]. Indigenous peoples are frequently categorized as minority groups due to their exclusion from regional political, security, and economic systems [3].

Health illiteracy has significant negative impacts on health outcomes, increasing the likelihood of incorrect health practices. One severe consequence of this is stunting. A child can experience stunting if they do not receive the minimum nutritional requirements needed for normal growth. Factors such as low health literacy, poverty, and poor surveillance systems can contribute to inadequate nutrition, leading to stunting [4].

Research indicates that Indigenous people are particularly vulnerable to stunting due to the combined effects of health illiteracy, poverty, and insufficient surveillance within their communities [5,6]. Studies of various Indigenous populations, such as the First Nations in Canada [7] and the Aboriginal and Torres Strait Islanders in Australia [8], demonstrate that health literacy is a crucial predictor of health quality. However, Indigenous communities often face cultural, language, and policy barriers, as well as economic isolation and inequality, which make them difficult to reach through modern health promotion and education efforts [9].

In Indonesia, the Baduy community from Banten province is one such Indigenous group facing challenges related to health literacy and stunting. The Baduy people live in Kanekes, a mountainous village in the centre of the province, about 100 km from the provincial capital, Serang, and 150 km from Tangerang City, which is part of the Greater Jakarta area. The map below shows the location of the Baduy community.

Literacy is generally a person’s ability to read and write [10]. However, the definition of health literacy is more complex. Health literacy is “a broad range of skills and competencies that individuals develop to search for, understand, evaluate, and use health information and concepts to make informed choices, reduce health risks, and improve quality of life” [10]. Limited health literacy is a silent killer because it secretly causes individual deaths and economic damage in society. In line with this, the benefits of health literacy reach all life activities, such as household, work, society, and culture.

The level of health literacy is influenced by language, culture, and social capital factors [10]. Media use is also associated with health literacy because media is closely related to development, health behaviour, and literacy in general. Together with the education system and health system, as well as the influence of family and peers, mass media affects health literacy through changes in individual characteristics, such as language skills, culture, education, social skills, cognitive skills, physical abilities, and media use. Health literacy then determines health behaviour, health costs, and individuals’ use of health services [10].

Political or civil literacy also plays a crucial role in health literacy, enabling citizens to become aware of health issues and engage in decision-making through civic and social channels [10]. Health communicators can be individuals experiencing health problems or those who feel it is important to communicate health information to others. These individuals face challenges, such as difficulty explaining their illness, asking questions, and feeling embarrassed about their health issues [11].

An exploratory study on health communication with African American prostate cancer survivors and their families [12] revealed that the health issue that is easiest to communicate is the issue of death, followed by the issue of illness. They ranked six health topics from most to least avoided, and death was the least avoided issue. Issues that are less easy to communicate are the health problems in social relationships and the financial aspects of health. The issues that are most difficult to communicate are sexual and marital health issues, as well as health for non-heterosexual groups.

Health communication often starts within the family, with women typically being more approachable for health-related discussions than men. As a result, women frequently take on the role of health communicators or caregivers within families [12]. Even though reliable information is available from the government [13], the initial and most frequent health communication occurs at the family level, where health concerns are discussed, information is shared, and healthcare decisions are made.

From the perspective of non-sufferers, health workers are often highlighted as key communicators. Effective health communication skills in health workers can ensure that patients or the public understand necessary health information, address feelings of uncertainty, and build healing relationships [14].

In Indonesia, health policy defines stunting as impaired growth and development due to chronic malnutrition and recurrent infections, characterized by body length or height below set standards [15]. Stunting is widespread in the Baduy community. A 2009 report by Anwar and Riyadi [16] indicated that the prevalence of stunting in the Baduy community was 60.6%, much higher than the national average at that time, which was only 36.8%. Meanwhile, data from 2022 showed that the prevalence of stunting in this community was 54% [17], which shows a low decrease when considered at the country level. The stunting prevalence in Indonesia is already down to 21.6% [18].

This research takes a health communication approach to address health literacy and stunting in the Baduy community. The study aims to determine the factors, impacts, and solutions for health illiteracy and stunting in the Outer Baduy of Kanekes Village. The article proceeds as follows. Section 2 describes the sociocultural characteristics of the Outer Baduy community. Section 3 presents the qualitative research methodology used in this exploratory study. Section 4 highlights the empirical findings related to stunting literacy. Finally, Section 5 discusses the research implications for academics and practitioners, focusing on Indigenous communities such as the Baduy.

## 2. Outer Baduy Community

Culturally, Baduy society is divided into Inner Baduy (*Baduy Dalam*) and Outer Baduy (*Baduy Luar*). The Inner Baduy community is classified as “earth hermits”, a group of people who adhere strongly to conservative principles by not changing their way of life over time and not having any contact with outsiders. Natural resources and the environment are carefully preserved because they are the only mainstay of people’s lives [19]. There are predictions that the Baduy people, especially the Inner Baduy, will soon become extinct due to being too strict and not opening themselves up to the outside world. Meanwhile, the Outer Baduy community is a community that still strictly maintains its customs but has opened up a little space for transactional relationships and does not lose its customs with outside communities [19].

As an Indigenous society, the Baduy people hold various types of taboos. Taboos in Baduy society are of three types, namely to protect the soul (human), to safeguard the purity of the *mandala* (motherland), and to preserve the purity of traditions. An example of a taboo to ensure the purity of tradition is the script taboo, namely the prohibition against learning Latin and Arabic letters. Old Sundanese letters are still permitted, and several *tangtu* (traditional leaders) know *cacarakan* (Sundanese alphabet) [20].

According to the Baduy’s beliefs, their lifestyle is “asceticism in *nagara* (country), asceticism in the kingdom, asceticism in the *mandala* (motherland), and asceticism in the holy land (their village) [21]”. They lived ascetically, which meant farming according to their ancestors’ rules. The *puun* (chief spiritual leader) ensures the community adheres to these rules. There is one *puun* for each village in Inner Baduy. So, there is no single leader, only these three *puuns*.

The Outer Baduy area does not surround the geographical area of Inner Baduy. The southern area of Kanekes village directly connects the Inner Baduy area with the Outer Baduy area without going through the Outer Baduy area. However, this area is naturally challenging to penetrate because it is hilly, with two hills, namely Kendeng and Hoe. The Inner Baduy area includes the villages of Cikartawana, Cibeo, and Cikeusik.

However, the geographical areas with exact boundaries, like today, are only approximate and can change over time. This change is because the villages in Kanekes Village can change position for various reasons. A village can move because of a fire that damaged many wooden houses or because of an idea from their ancestors to spiritual leaders who asked them to move [22].

Most of the Baduy people live as farmers. However, the Indonesian government has made Kanekes a cultural tourism destination so that there is additional financial income for the Baduy community from this sector. Apart from that, many visitors from the outside come to see the life of the Baduy people. They are also widely known as one of the ethnic groups in Indonesia that still adhere to solid customs and traditions. The Baduy people refuse to be called an isolated community and have committed to planning their future with the government and outside communities [22].

The Baduy community, like Indigenous communities in other areas of Banten, has relations with the local government. For traditional leaders, annual ceremonies, such as *seba* (walking together to visit the government official in the city), are an effort to maintain and preserve the community’s cultural identity and a strategic communication medium for dialogue between Indigenous communities and the government [23].

## 3. Research Methods

### 3.1. Design

This research employs a case-study approach, a qualitative method where the researcher investigates a specific system (case) or multiple systems (cases) over time through in-depth data collection from various sources [24,25]. The purpose is to provide a detailed understanding of the factors, impacts, and solutions related to health illiteracy and stunting in the Outer Baduy of Kanekes Village.

Given its objectives, this research falls under exploratory case-study research [26]. It focuses solely on the Outer Baduy community without comparing other cases, so it qualifies as a single case study [27]. Furthermore, this study is intrinsic and was selected based on the case’s uniqueness rather than its representativeness or comparability with other cases [26]

### 3.2. Participants and Setting

The setting for this research is the Baduy community in Kanekes Village, specifically in the hamlets of Ciranji Pasir, Ciranji Lebak, Cijanar, Ciemes, and Cibagelut. These locations were chosen because they are near the village health post, managed by the Indonesian Volunteer Friends (SRI), an NGO. This health post, established on 20 November 2021, replaced a defunct government health post and continues to detect many stunting cases in the area (see Figure 1).

Six participants were selected using a snowball sampling technique. Key individuals were initially identified and then referred to others for interviews. A key individual in this research was a midwife from the Dompet Dhuafa Foundation who worked as a volunteer to serve the Baduy people. She referred four people, including the head of the NGO SRI, a midwife from Dompet Dhuafa, and two mothers who have stunted children from the Baduy community. The head of the NGO SRI referred to the head of the Lebak Regency social service, while the local mother referred to a community’s neighbourhood leader and another mother. According to Creswell [24], this purposive sampling technique identifies rich-information cases from people with specific knowledge or experience relevant to the phenomenon under study—in this case, the factors, impacts, and solutions related to health illiteracy and stunting in the Outer Baduy of Kanekes Village.

A brief analysis of the interview data revealed that theoretical saturation was achieved among six participants. No new information emerged from the seventh participant, another mother of stunted children, indicating that further data collection would not yield new insights, thus concluding the data-collection process.

### 3.3. Data Collection

Data-collection techniques included semi-structured interviews and secondary data analysis. Semi-structured interviews allowed for a directed exploration of causal chains in the case study. Three groups of informants were interviewed: Baduy community members (including two mothers with stunted children, referred to as Baduy 1 and Baduy 2), one Baduy community leader (at the neighbourhood level), and external stakeholders (an NGO officer, a midwife, and a government social service officer). The interviews covered factors associated with low health literacy, its impact, and strategies to improve health literacy (Table 1).

Secondary data were obtained from a YouTube video titled “Multidisciplinary Talkshow: Stunting in Baduy”, [28] featuring researchers and a health worker who provided services to prevent stunting in the Baduy community. Relevant researchers included one archaeologist, one dentist, and one nurse. The Directorate of Community Service and Empowerment, University of Indonesia (DPPM—UI), created the video to document their community service program.

### 3.4. Ethical Considerations

The participants were informed about all study activities, and informed consent was obtained before data collection. The participants were assured of their right to request additional information and were told that their data would be confidential and accessible only to the researcher. An approval letter was obtained from the Padjajaran University Research Ethics Committee (KEP UNPAD) for performing research on the Baduy community.

### 3.5. Data Analysis

Data analysis was conducted using a holistic approach, which involves a comprehensive examination of the entire case or certain aspects [24]. This research employed holistic analysis through two main steps, namely description and thematization. The case was described in detail, from its history to daily activities, and critical issues were identified to gain a deeper understanding of the case’s complexity. Themes were developed from transcriptions and coded interview texts.

The research addressed dependability, confirmability, transferability, and credibility to maintain quality [29]. Dependability is addressed through method triangulation, using interviews and video to verify data consistency. Confirmability is achieved through confirmability audits, re-checking data, and analysis results for consistency and neutrality. Transferability is how generalizable research results are in other contexts [30]. Transferability is addressed by providing a detailed description of contextual factors, enabling comparisons. Finally, credibility is maintained through source triangulation, comparing interview results from different sources and member-testing methods to accurately interpret participants’ responses [31]. Member testing is conducted by accurately interpreting the interviewee’s reactions during the research process [24].

## 4. Results

First, we will describe the factors associated with the low health literacy of the Outer Baduy community and its impact on this community. Second, we present the results of a qualitative analysis that produces five themes that describe the strategies that stakeholders can use to overcome health literacy in the Outer Baduy community, including developing the health literacy of community leaders, managing information-technology-based health-information groups, always having at least one health worker present in the community, encouraging joint reflection when health cases occur, and balancing gender communication. The table below summarises the analysis’s themes, subthemes, and categories. The data were triangulated by comparing the interview and the video. We found no conflicting information. Both sources complement each other.

### 4.1. Factors Associated with Low Health Literacy in the Baduy Community

The Baduy people’s health illiteracy is closely related to broader illiteracy due to their inability to read. The role of this general illiteracy is revealed in the narrative of the Baduy mother herself when she had to ask the midwife repeatedly about the writing on the family planning acceptor card:

“*If I do not repeat it, I was afraid I will not understand the writing on the card, so I ask Teh Ira*”.(Baduy 2)

SRI volunteers revealed that customs regarding the prohibition of eating certain foods were a limiting factor because some of the prohibited foods, such as beef and chicken, were quite nutritious. This prohibition discourages people from seeking health information related to nutrition and its role in growth and health:

“*We use Baduy as the stunting pilot project site because this tribe emphasizes many prohibitions related to food, related to customs that must not be violated*”.(NGO)

The strictness of this prohibition was also expressed by a source from the Lebak Social Service who described the strict prohibition regarding food so that people refused large amounts of additional food provided by the government:

“*At that time, the Ministry of Social Affairs had food-giving activities in Baduy. We warned that Baduy people only need rice, salted fish, and shrimp paste. Nevertheless, they sent all other foods. Hence, many containers of food came just to be rejected by the Baduy at that time*”.(The Government)

A solution to the food problem is to give eggs to the Baduy family. Eggs are a good protein source and are not prohibited by customary law. However, the condition of the families was so bad that the eggs, originally provided for the children, were consumed by their parents:

“*We are ashamed. [The egg] should have been provided by their parents. We are also ashamed because the eggs are provided for stunted children, but their parents eat them*”.(Punggawa)

From the time dimension, Baduy people have difficulty increasing literacy because they spend time in the fields, both the husband and the wife. Midwife Ira revealed that she could not provide education to the community because the community was not at home. They work in the fields for their daily food needs:

“*Well, the target does not like anything like that, ma’am. For example, I want to provide additional food like that, giving eggs or holding a gathering for education and providing additional food like that. The target is in the huma [rice field], in the fields, so there are only 1 or 2. So, it is less effective here because of the people. They do not stay home every day, so they are also busy with their farming activities. Their livelihood is farming, right, ma’am? If they do not farm, they cannot eat because they earn money there; that is how it is*”.(Midwife)

Midwife Ira revealed that, in general, people learn by imitating what their parents did before rather than finding out health information about whether an action is appropriate or not:

“*And also looking at the elders’ previous [bad] habits, yes, they have been followed [uncritically by the community], so that is [a sign of] a lack of education, huh*”.(Midwife)

Meanwhile, a professor of archaeology at the University of Indonesia who took part in community service activities for the Baduy community emphasized that access is one of the factors associated with the low health literacy of the Baduy community. The Inner Baduy community is difficult to access, so health-literacy programs cannot be implemented. At the same time, the Outer Baduy community is more easily accessible, so the community is also more literate about health:

“*Well, what is interesting about our survey findings is that we divided three areas so there is Inner Baduy, and this is the area where access is difficult, so if we look for it, it is also problematic; there is also little knowledge about health and stunting. That is the characteristic of what is Inner Baduy. However, for Outer Baduy, communication is accessible because their knowledge regarding various kinds of programs has been widely accepted. On the other hand, because it is easy to access, many government, private, or community programs are carried out in the village. This village, so yes, is in the villages of Kaduketuk and then Cijahe and the surrounding areas, and the people are already used to receiving programs*”.(Archaeologist)

Another external factor is the lack of consistency from external parties in increasing the literacy of the Baduy community. The programs provided are unsustainable, so they are less effective in increasing community literacy. The head of one neighbourhood (*punggawa*) said:

“*Someone once told about stunting, but there was no further follow-up*”.(Punggawa)

Finally, another interesting cultural factor is gender segregation. The researcher questions the understanding of the problem by asking if it is understood. Baduy 1 responds, indicating a lack of understanding and suggesting that men might understand it better:

People usually come here from the health centre or even the health department. They are people who are far away. Did they give health information? (The Author)

*No, maybe men understand*.(Baduy 1)

Baduy 1 insists that health information, especially about stunting, should be conveyed to men for clarity:

“*If you talk like this [health information about stunting], if you want it to be clear, you have to convey it to a man*”.(Baduy 1)

When asked why health information should be communicated to men, Baduy 1 expresses uncertainty about the exact reason. Baduy 1 attributes this preference to traditional gender roles and practices, mentioning that men typically handle such matters and that community meetings are attended exclusively by men:

“*Don’t know. That’s a man’s business. They held a meeting together. You should talk about this with a man. If there is a community meeting here, all the people who come are men*”.(Baduy 1)

The dialogue reflects cultural norms and gender roles within the Baduy community, where men are seen as the primary recipients and conveyors of important information. It highlights a potential barrier to effective health communication, as women might not be seen as appropriate recipients of health-related information despite their critical role in child rearing and health maintenance.

### 4.2. The Impact of Low Health Literacy in the Baduy Community

The most pronounced impact of the low health literacy of the Baduy community is fatalism. Society views a health condition as destined, and it must be accepted without being corrected. For example, when asked why a mother’s child is stunted, the reason given is genetic rather than nutritional problems:

“*His father’s [body] was small, so Sardin’s [body] was small*”.(Baduy 1)

The fatalism of the Baduy people does not appear to be ideological, at least in terms of health, as revealed in the case of snake bites. The Baduy community understands that snakes are their greatest threat. Despite knowing the danger, they cannot avoid it because their work involves being in snake-prone areas. They feel resigned to their situation due to the lack of alternatives. So, there is a philosophical acceptance of death as something predestined. However, the introduction of medical intervention brings hope for improvement.

“*So far, they have completely ignored the data regarding deaths due to snake bites, thinking that they have just given up. This belief [the folk legend that Baduy is immune to snake venom] is just a tactic from kokolot [community elders]. [he actual belief is that] when someone’s life is ended [by snake bites], it is predestined. There’s something [belief] like that. But when medical people come in and all that [health facilities], [they start to believe that] there are things that can be fixed*”.(NGO)

Another impact is the high maternal and child mortality rate. This fact was revealed by a resource person from the University of Indonesia Hospital in a community service talk show:

“*Well, yesterday, we looked at the data from the last few years. Yes, in the last two years and three years, Baduy had a high maternal mortality rate. Yes, up to 4 people per year. Well, this is quite extraordinary. Just one is quite high. Here, it is up to four every year. The child death figure is even higher. There are cases of child deaths, especially neonates. They died when they were born. Well, this case is also quite high. Between 9 and 14 children per year die in Baduy*”.(Nurse)

Lack of health literacy also puts the Baduy community in a vicious circle. Low health literacy makes them less able to maintain their health and protect themselves from disease. This condition leads to health costs that they cannot afford. Ultimately, because of this inability, they become even more fatalistic, as shown in an interview with a Baduy’s neighbourhood leader:

“*Sometimes there are things like this, ma’am. For example, the village used to be a tiny population. Even then, they said no matter how badly my child is sick, do not take him to the medical centre. Could you not take him to Rangkasbitung? It is expensive, so where do we get the money to pay the bill? That is the chatter of old people is like*”.(Punggawa)

### 4.3. Strategies to Improve Baduy Community Health Literacy

#### 4.3.1. Developing the Health Literacy of Community Leaders

The Baduy community still depends on leaders like *Jaro* and other traditional authorities. So, if community leaders have sufficient health literacy, they can pass it on to other community members. So far, community leaders have encouraged public health, but more aspects of health actions have not yet reached the health-literacy stage:

“*The point is that Jaro and Puun also urge the Baduy people to consume nutritious food, but not force it because there is a clash of customs [such as eating healthy but prohibited food] that cannot be forced*”.(Midwife)

The potential for Jaro to become a community literacy agent in the health sector is enormous because the community attends monthly meetings to discuss all matters related to customary law. In this event, recommendations for health literacy can be included:

“*We were invited to a community gathering every month. The purpose of the meeting is to provide instructions to the village or the parents; everything is talked about and entrusted to the parents. Only recently has this happened. Violations of various kinds are reported, and then there are instructions, such as if you [want to] eat [certain] food [from outside], you must be careful [that the food contains prohibited or unhealthy items]. It was recently held with the community. Every month, the meeting is held*”.(Baduy 1)

#### 4.3.2. Managing Information-Technology-Based Health-Information Groups

Even though the Baduy people live traditionally, they still have cell phones and use them mainly for business purposes [32]. One of the central aspects of health literacy is the ability to search for and sort information using information technology. Baduy people are not used to looking for health information using their cell phones. However, according to customs, information technology is only limited to the Outer Baduy area:

“*Technology can come in slowly. Cell phones are already in, but if they want to use them, they must go to Outer Baduy, ma’am. In Inner Baduy, they must go outside the village first. Then they are now free to meet their friends*”.(Dentist)

So far, many Baduy already have the cellphone numbers of health workers.

“*I have all [the midwives’ cellphone numbers]. I have the [cellphone] number of the health personnel and midwives in Cibaleger, in Kariki, I have all the [cellphone] numbers*”.(Baduy 1)

However, health workers still tend to be passive. With access to contact with health workers, a WhatsApp group containing people from one village with specific healthcare workers can be formed. These health workers provide regular and contextual information to increase public health literacy.

#### 4.3.3. Always Present at Least One Health Worker among Residents and Provide an Example of Healthy Living

The visual presence of a health worker among community members emphasizes the importance of health aspects and creates a sense of security for the community. The presence of health workers in the community is something that the Baduy people themselves want:

“*Indeed, in the past, I heard that people from Baduy Dalam, for example, were not allowed to use medical services because it has been like that for generations. Nevertheless, now we live in a wider society. So, people from Outer Baduyhave already used modern medicine, right? So sometimes what else can we do if we do not get help*”.(Punggawa)

Community health workers are also vital to help the community overcome health problems. The following presentation from the SRI chairman shows the importance of the presence of health workers:

“*My intention here is nothing different. I do not sell medicine. I do not sell these or those. I want to help the Baduy people because they do not know where to go. Mr. Jaro said, “Do not leave the Pustu [Subsidiary Health Center]”. No, I will not leave it. If the condition is like this, then that is the condition. So, in the end, that is it. Yesterday, Mr. Jaro’s wife had her feet scalded with hot water. I have taken it to the doctor, there are burns. Then, given ointment, she is healthy now. “It was Mr. Jaro who asked for treatment directly*”.(NGO)

#### 4.3.4. Encouraging Collective Reflection When Health Cases Occur

In extreme situations, it is essential to reflect so that the same incident does not happen again. Incidents in the form of deaths due to health problems can be raised in communications, as long as they remain sensitive to customary law and ethics, as learning material for the community to be more careful and maintain their health. For example, the incidence of breast cancer can be used as a lesson to avoid foods that are not nutritious and contain carcinogenic substances:

“*That is why I say when I go around, “Do not eat cilok [chewy tapioca balls], do not eat noodles, better eat boiled bananas”. This local wisdom has now been lost. The children eat cilok every morning; there are cilok around. Snacks. That afternoon, a lady on a motorbike was picked up by her husband and shouted “cilok…cilok”. The children do not want to eat dinner anymore; why? Because the taste is different, there is already flavouring, and all kinds of things, and a generation of flavouring-addicted people has emerged*”.(NGO)

The research team from the University of Indonesia also revealed an incident of delay in providing assistance, which resulted in the death of a mother.

“*For example, there was a risky pregnancy, but it turns out that because of a customary problem, she was not allowed to access health services. Finally, after negotiating with the traditional authorities, she was allowed. However, 15 min before arriving at the hospital, the mother died. The mother died on the road. So she died in the ambulance*”.(Nurse)

A wise and sensitive approach to customary law is needed well in advance to increase public health literacy so that the same incident does not happen again.

#### 4.3.5. Balancing Gender Communication

Baduy culture places the male gender as the gender that deals with governmental, administrative, and technical aspects. In contrast, the female gender still plays the traditional role of managing the household and family. Because health literacy does not look at gender, both genders must receive an education that is balanced and adapted to gender. This gender difference is also observed in various locations worldwide, such as Ghana [33] and Hong Kong [34], where men generally have higher health literacy than women. An efficient gender-specific approach needs to be developed to improve the health literacy of the Baduy community [35]. For example, health literacy related to nutrition can be directed at women because they culturally have an essential role in agriculture [36]. In contrast, procedural and heuristic literacy can be directed at men.

## 5. Discussion

The research results reveal that several factors contribute to the low health literacy of the Baduy community. These factors include low general literacy (reading and writing), customs that impose prohibitions on eating certain foods, spending extensive time in the fields, learning through imitation of parental behaviour, and difficulties in physical access. Additionally, from outside the village, inconsistencies in health-campaign programs and gender segregation further exacerbate the issue.

These findings align with the existing literature that highlights the challenges disadvantaged community groups face in improving health literacy [37,38]. For example, Oredola et al. [39] identify that Indigenous communities worldwide often transmit knowledge through deliberate instructions from parents to children, with children learning by imitating their parents’ behaviour, making parents key to health promotion among their children. In Indigenous Australian adults, geographical remoteness has been identified as a significant barrier to health literacy [40].

However, this research presents some differences compared to studies conducted on Aboriginal communities in Australia. For instance, Nash and Arora [41] reviewed studies that did not mention eating taboos as a constraint. Consequently, many interventions aimed at Aboriginal communities focus on nutritional aspects, such as lowering food prices and increasing food availability. In contrast, the situation is more complex for the Baduy community, which adheres to several food taboos. Unlike other Indigenous communities, which typically have food taboos only during specific occasions such as pregnancy and postpartum [42], the Baduy community restricts their food choices at all times and under all conditions. This practice aligns with their philosophy of living a simple life, which also aligns with their low socio-economic status and benefits them economically.

The existing obstacles are deeply connected to the low socio-economic status of the Baduy community. Extreme poverty, in addition to strict cultural practices, plays a role in hindering the development of health literacy. For instance, the taboos on eating certain foods and the extensive time spent working in the fields reflect the community’s low-income levels, compelling them to forgo high-protein foodstuffs and focus on increasing productivity through intense labour.

Interviews with community members also revealed several impacts of low health literacy, such as fatalism, maternal and child deaths, and issues related to health costs. These problems could be mitigated if the Baduy people had higher health literacy. High health literacy empowers individuals to seek health assistance, anticipate factors that increase the risk of maternal and child mortality, and take preventive steps to manage health costs effectively.

Several strategies for increasing health literacy have been proposed to address these challenges. These include developing the health literacy of community leaders, managing information-technology-based health-information groups, ensuring the presence of at least one health worker among residents, encouraging joint reflection when health cases occur, and balancing gender communication. These strategies align with the previous literature on improving health literacy, which emphasizes context-sensitive interventions that address significant socio-economic and emotional challenges [43,44,45,46,47]. The proposed interventions focus on developing the health literacy of community leaders, managing health information through technology, ensuring consistent health worker presence, fostering community reflection on health issues, and promoting balanced gender communication.

### Limitations

We checked the data and analysis results again to maintain confirmability. The transferability of this research into another culturally shocked Indigenous population might be possible, given the similarity in their contextual factors such as history and geography. The credibility of this research was maintained because the participants also read the transcript. However, the limited number of interviews and participants in this research indicates a constraint of the study. Although this is common in case-study research, the small sample size prevents broader generalizations. Future studies should aim to include a more significant number of participants to enhance the robustness of their findings. It is also important to note that the first author, who conducted the interviews, was not fluent in Baduy Sundanese, the language spoken by the Baduy community. Consequently, a midwife from the Dompet Dhuafa NGO, who participated in this research, was chosen as the translator. This decision may have introduced bias into the data-collection process.

The researcher’s role in this research is participative, in the sense that we participate in helping the community reach more health. We joined the village midwife to tour the community and assist in any health intervention she did for the community members during the research. The involvement is the strength of this research, in the sense that we could experience the problems that the community faced. However, this also introduces bias, especially for theoretical saturation. We reached saturation after mothers from the community kept providing the same information, especially regarding male–female health literacy. If we asked more community leaders, the saturation point might be further, and the data might be richer.

Future research could examine the gender dynamics within the Baduy community and how they impact health communication and literacy. Research should look into effective ways to address gender-specific health needs and communication barriers. Another venue for future studies is the study on the feasibility and impact of using information technology, such as mobile apps, WhatsApp groups, and YouTube channels, for health education. This research should explore the community’s access to and engagement with these technologies. Another study might explore the effectiveness of mobile health clinics for providing health services and education to the Baduy community and explore various health-communication strategies to identify the most effective methods for reaching and educating the Baduy community.

## 6. Conclusions

For a long time, the Baduy people in Indonesia have lived by adhering to their customs and maintaining strict isolation from the outside world. However, as access to the outside world increases and technological advances develop, many health problems in Baduy society must be addressed immediately.

We, as outsiders, recognise the influence and power of customary law in determining the lives of the Baduy people. We also realise that the Baduy people have the right to health and should have the broadest possible access to health services, coupled with increased health literacy, which empowers them to maximise available health facilities. Addressing this issue involves many stakeholders, and their alignment is complex and requires precise and intense communication.

In this research, six sources discussed the factors, impacts, and ways to overcome low health literacy in the Baduy community. This research found that general illiteracy in the Baduy community, dietary restrictions, livelihoods, learning methods, and gender segregation play an essential role in the low health literacy of the Baduy community. Difficult access to villages and the lack of consistency of external parties in providing health programs also play a role in health literacy. Mentally, this has the impact of fatalism in the society. Physically, health literacy affects the high maternal and child mortality rates and high health costs. The strategy proposed to increase the health literacy of the Baduy community based on the findings of this research is to encourage collective reflection when extreme health cases occur in the community. An emphasis on developing health literacy that targets community leaders is also crucial because the Baduy community generally respects community leaders.

Furthermore, balancing gender communication is a solution that suits the existing situation of gender segregation. The fact that people have low mobility because they are always in the village and are not allowed to use vehicles demands the presence of at least one health worker among the residents who provides an example of healthy living through direct communication. Managing information-technology-based health-information groups is essential, and this research suggests establishing WhatsApp groups and YouTube channels that socialise and teach health literacy to the Baduy community.

The findings indicate that several factors, including general illiteracy, cultural practices, and time constraints, drive low health literacy in the Baduy community. Consistent and long-term health programs are crucial for gradually overcoming these deep-rooted issues. Short-term interventions are unlikely to produce lasting change because they do not allow enough time to build trust, adapt to cultural nuances, or make significant improvements in literacy and health knowledge.

Furthermore, the impacts of low health literacy, such as fatalism, high maternal and child mortality, and a vicious cycle of poor health maintenance, are severe and complex. These issues cannot be resolved with temporary solutions; they require ongoing education, support, and adaptation of health programs to the community’s evolving needs. Long-term partnerships can help build a resilient health infrastructure that continuously addresses these challenges.

Hence, as a policy recommendation, we advise that external health programs are consistent and sustained. The government must avoid short-term interventions and instead focus on long-term partnerships with NGOs, government agencies, and other stakeholders to provide ongoing support and resources for the Baduy community. The government needs to use the partnership to implement regular health assessments to monitor the health status of the Baduy community and identify emerging health issues. After that, the partnership should use these data to adapt and improve health programs continually. In line with this strategy, the partnership should train and employ local Baduy community members as health workers. These individuals can liaise between the health system and the community, providing education and basic health services while respecting cultural norms. Meanwhile, the partnership can design health programs that address gender segregation by providing separate but equal health-education sessions for men and women. Ensure that both genders receive the same quality and quantity of health information.

## Figures and Tables

**Figure 1 ijerph-21-01114-f001:**
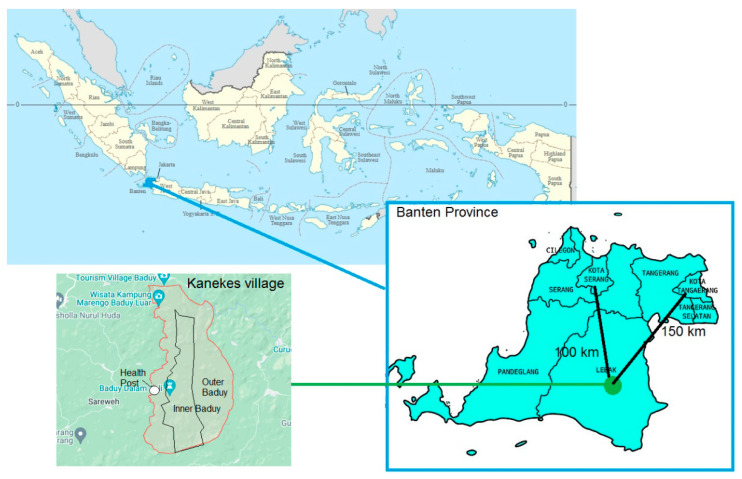
The location of the Baduy community. Map source: Wikipedia and Google Maps, CC BY-SA 3.0.

**Table 1 ijerph-21-01114-t001:** Themes, subthemes, and categories from the analysis.

Themes	Subthemes	Categories
Factors associated with Low Health Literacy	General Illiteracy	Inability to read and dependence on verbal explanations
	Cultural Practices	Prohibitions on certain nutritious foods
	Accessibility	Remote locations and difficult access for Inner Baduy
	Time Constraints	Community members busy with farming activities
	Learning Methods	Imitating elders rather than seeking new health information
	Program Consistency	Lack of follow-up on health programs
	Gender Segregation	Different expectations for health literacy between men and women
Impact of Low Health Literacy	Fatalism	Acceptance of health conditions as destiny
	High Maternal and Child Mortality	High rates of maternal and neonatal deaths
	Vicious Circle	Low health literacy leads to poor health maintenance, higher health costs, and further fatalism.
Strategies to Improve Health Literacy	Developing the Health Literacy of Community Leaders	Empowering leaders like Jaro and traditional authorities to educate the community
	Managing IT-Based Health-Information Groups	Utilising cell phones and WhatsApp groups for health-information dissemination
	Presence of Health Workers	Ensuring at least one health worker is always present in the community
	Encouraging Collective Reflection	Discussing health incidents to learn and prevent future occurrences
	Balancing Gender Communication	Providing gender-specific health education: nutrition for women and procedural literacy for men

## Data Availability

The original contributions presented in the study are included in the article, further inquiries can be directed to the corresponding author.

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
