# Peer review of "Addressing Health Illiteracy and Stunting in Culture-Shocked Indigenous Populations: A Case Study of Outer Baduy in Indonesia"

_ijerph, 2024, doi:10.3390/ijerph21091114_

Round 1
Reviewer 1 Report (Previous Reviewer 1)
Comments and Suggestions for Authors
I appreciate the authors for their time and efforts to revise the manuscript. Although I didn’t have access to the point-by-point reply to the comments and I think the authors have not uploaded the file, I can see the manuscript has significantly improved based on the revision. There is just one minor issue. Please remove any names or identifications from the quotes and present them anonymously. The authors can use numbers for the interviewees, e.g., Interviewee 1, nurse 6, etc.
Author Response
Thank you for your thoughtful feedback and for recognizing the improvements in the manuscript. We apologize for the oversight in not uploading the point-by-point reply, and we appreciate your patience. We have taken your suggestion on board and have now anonymized the quotes by replacing names with stakeholder type: Baduy 1, Baduy 2, Punggawa (small community leader), NGO, midwife, government, archaeologist, dentist, and nurse." We hope this revision meets your expectations.
Reviewer 2 Report (Previous Reviewer 2)
Comments and Suggestions for Authors Congratulations to the authors for a much improved manuscript in terms of readability and clarity in the methods and conclusions. However, I feel that further improvements are needed - there are still many sentences which are not clear or have an ambiguous message, for example in the introduction "Many countries face high levels of health illiteracy and prevent health campaigns from perpetuating already existing illiteracy[1]." Do authors really want to state that many countries prevent health campaigns from perpetuating illiteracy? Probably the foreseen message is a different one. Similarly, stunting is not necessarily caused by the combination of deficient nutrition and surveillance... ("A child can experience stunting if health literacy is combined with poverty and a poor surveillance system, so the child and mother can not meet the minimum nutritional requirements needed for average child growth") - the sentence seems to imply that for stunting health literacy should be combined with poverty and poor surveillance... but really what causes the stunting is the lack of meeting the minimum nutritional requirements... even rich children will get stunted if they do not receive the nutrition they need. Of course, often these external factors lead to not reaching the minimum nutritional requirements but people with (low or high) health literacy, poverty and poor surveillance may or may not experience stunting.... Page 5 lines 189-192 please mention only the prevalence, absolute numbers are not so informative without the population sizes - it would be useful to also know the national prevalence of stunting in Indonesia as a comparison. At the end of the introduction the authors state "This research takes a health communication approach to address health literacy and stunting in the Baduy community, focusing on the communication issues between public health stakeholders and the Baduy community. The study aims to determine the factors, impacts, and solutions for health illiteracy and stunting in the Outer Baduy of Kanekes Village." However, in chapter 3 (methods) it is stated that "The purpose is to provide a detailed case description and identify case-based themes [22]. This method is chosen because it connects abstract ideas with concrete cases observed in detail. Here, the abstract idea is a stunting communication strategy, while the concrete case is its implementation in the real world." Then, later on, authors state that it is aimed at developing a hypothesis or theory and in the next sentence that the aim is to create a cross-cultural communication model. Those are 4 different objectives written with few words separating each. From the results, where factors related with low literacy, five themes identified for strategies to overcome health literacy - it is NOT an evaluation of the stunting communication strategy but rather information that may inform such a communication strategy? Is that right? So please restructure your objectives/aims and methods to what was really presented in the paper and if this is indeed the objective, begin the discussion with how the gathered information will help in designing the communication strategy. Or perhaps the objective was to identify factors associated with low literacy or with stunting (two different concepts!, as it seems to be from the first paragraph of the discussion. If that is the case, please organize objectives, methods and results accordingly. It is different to study qualitative communication issues versus the study of the implementation of a stunting communication strategy. I would presume that as the basis for the development of the communication strategy, a qualitative study on existing communication issues was done, but from the objectives and methods it is not clear what was really the objective of the study. Identifying the aim of the work presented in this particular paper and the results of studying the data collected with that aim will help structuring the paper and conveying the key message. For example, in 3.2 it is not clear with which aim the participants were invited - what were they told they were invited to, were they introduced to the stunting communication strategy or not? Maybe this can be of help in identifying this aim. Similar for 3.3 in data collection. 4. The results begin with triangulation without describing the results - the part of triangulating would be better placed a bit later in this chapter of the paper. Considerations regarding transferability/generalizability should be in the discussion, not the results section. Discussion: align with what is decided as the main objective and in the part of policy recommendation, make sure that you explain why you come to these recommendations. Some seem a bit disconnected to the findings of your study - which may again be related with the disalignment between the many objectives, results and conclusions. Improving the "backbone" of the study in terms of the aims and rewriting the whole document according to that backbone will help avoiding this issue.Comments on the Quality of English Language
The structure of sentences and paragraphs are often confusing, leaving space for multiple interpretations. English in itself is fine.
Author Response
|
Comments 1: Congratulations to the authors for a much improved manuscript in terms of readability and clarity in the methods and conclusions. However, I feel that further improvements are needed - there are still many sentences which are not clear or have an ambiguous message, for example in the introduction "Many countries face high levels of health illiteracy and prevent health campaigns from perpetuating already existing illiteracy[1]." Do authors really want to state that many countries prevent health campaigns from perpetuating illiteracy? Probably the foreseen message is a different one
|
|
Response 1: Thank you for your constructive feedback and for acknowledging the improvements in the manuscript. We understand your concern about the clarity of the sentence in the introduction. Based on your suggestion, we have revised the sentence to better convey the intended message: "Many countries struggle with high levels of health illiteracy and low effectiveness of health campaigns for their citizens." We believe this revision more accurately reflects the issue at hand and appreciate your input in helping us enhance the clarity of the manuscript.
|
|
Comments 2: Similarly, stunting is not necessarily caused by the combination of deficient nutrition and surveillance... ("A child can experience stunting if health literacy is combined with poverty and a poor surveillance system, so the child and mother can not meet the minimum nutritional requirements needed for average child growth") - the sentence seems to imply that for stunting health literacy should be combined with poverty and poor surveillance... but really what causes the stunting is the lack of meeting the minimum nutritional requirements... even rich children will get stunted if they do not receive the nutrition they need. Of course, often these external factors lead to not reaching the minimum nutritional requirements but people with (low or high) health literacy, poverty and poor surveillance may or may not experience stunting....
|
|
Response 2: Ok. The sentence now changed into these sentences: “A child can experience stunting if they do not receive the minimum nutritional requirements needed for normal growth. Factors such as low health literacy, poverty, and poor surveillance systems can contribute to inadequate nutrition that leads to stunting”
|
|
Comments 3: Page 5 lines 189-192 please mention only the prevalence, absolute numbers are not so informative without the population sizes - it would be useful to also know the national prevalence of stunting in Indonesia as a comparison. |
Response 3:Agree. The sentence now changed to: “A 2009 report by Anwar and Riyadi[19] indicated that the prevalence of stunting in the Baduy community was 60.6%, much higher than the national average that time, which only 36.8%. Meanwhile, data from 2022 showed prevalence of stunting in this community was 54% (Dompet Dhuafa, private communication), which show low decrease, consider at the country level, stunting prevalence in Indonesia already down to 21.6% (Survai Status Gizi Indonesia, SSGI, 2022).”
|
|
Comment 4: At the end of the introduction the authors state "This research takes a health communication approach to address health literacy and stunting in the Baduy community, focusing on the communication issues between public health stakeholders and the Baduy community. The study aims to determine the factors, impacts, and solutions for health illiteracy and stunting in the Outer Baduy of Kanekes Village." However, in chapter 3 (methods) it is stated that "The purpose is to provide a detailed case description and identify case-based themes [22]. This method is chosen because it connects abstract ideas with concrete cases observed in detail. Here, the abstract idea is a stunting communication strategy, while the concrete case is its implementation in the real world." Then, later on, authors state that it is aimed at developing a hypothesis or theory and in the next sentence that the aim is to create a cross-cultural communication model. Those are 4 different objectives written with few words separating each. From the results, where factors related with low literacy, five themes identified for strategies to overcome health literacy - it is NOT an evaluation of the stunting communication strategy but rather information that may inform such a communication strategy? Is that right? So please restructure your objectives/aims and methods to what was really presented in the paper and if this is indeed the objective, begin the discussion with how the gathered information will help in designing the communication strategy. Or perhaps the objective was to identify factors associated with low literacy or with stunting (two different concepts!, as it seems to be from the first paragraph of the discussion. If that is the case, please organize objectives, methods and results accordingly. It is different to study qualitative communication issues versus the study of the implementation of a stunting communication strategy. I would presume that as the basis for the development of the communication strategy, a qualitative study on existing communication issues was done, but from the objectives and methods it is not clear what was really the objective of the study. Identifying the aim of the work presented in this particular paper and the results of studying the data collected with that aim will help structuring the paper and conveying the key message. For example, in 3.2 it is not clear with which aim the participants were invited - what were they told they were invited to, were they introduced to the stunting communication strategy or not? Maybe this can be of help in identifying this aim. Similar for 3.3 in data collection
|
|
Response 4: The aim of this paper is to provide a detailed understanding of the factors, impacts, and solutions related to health illiteracy and stunting in the Outer Baduy of Kanekes Village, as stated at the end of introduction section. What we mean as the detailed case description in chapter 3 is the general aim of case study method that we use. To make this clear, we correct the paragraph in the chapter 3 such that the same with the aim of this research. Then, “aimed at developing a hypothesis or theory” is the general aim of exploratory research. We erased the phrase to avoid confusion. We also erased the next sentence “Specifically, it seeks to create a cross-cultural communication model to address stunting in the Outer Baduy community.” We do not evaluating stunting communication strategy, so we erased this sentence : “This method is chosen because it connects abstract ideas with concrete cases observed in detail [26], allowing for a comprehensive analysis of the stunting communication strategy and its real-world implementation” and also this paragraph “Effective health communication strategies are essential to addressing these issues. The failure to improve stunting and general health in the Baduy community is often due to ineffective communication between stakeholders (national and local public health officials and health non-governmental organizations) and between stakeholders and the Baduy community. This poor communication leads to inefficiencies, a one-way flow of information, and mismatched customs and policy formulations. Most programs, such as malnutrition improvement and poverty eradication, are not sustained long enough to have a lasting impact.” Also this sentence erased “focusing on the communication issues between public health stakeholders and the Baduy community” And this sentence too “It contributes to communication science by introducing a framework for understanding communication processes within the Outer Baduy community and to public health by linking this framework to stunting.” We change the phenomenon in chapter 3 to “the factors, impacts, and solutions related to health illiteracy and stunting in the Outer Baduy of Kanekes Village.”
|
|
Comment 5: The results begin with triangulation without describing the results - the part of triangulating would be better placed a bit later in this chapter of the paper.
|
|
Response 5: Thank you. The results now put at the end of the paragraph that lead to Table 1
|
|
Comment 6: Considerations regarding transferability/generalizability should be in the discussion, not the results section.
|
|
Response 6: Thank you. The considerations regarding transferability/generalizability put at the beginning of Limitation section
|
|
Comment 7: Discussion: align with what is decided as the main objective and in the part of policy recommendation, make sure that you explain why you come to these recommendations. Some seem a bit disconnected to the findings of your study - which may again be related with the disalignment between the many objectives, results and conclusions. Improving the "backbone" of the study in terms of the aims and rewriting the whole document according to that backbone will help avoiding this issue.
|
|
Response 7: The discussion section actually the backbone of the study anchored to the aims to explore “the factors, impacts, and solutions related to health illiteracy and stunting in the Outer Baduy of Kanekes Village.” For the policy recommendation, we add two paragraphs that connect the finding with the recommendation: “The findings indicate that low health literacy in the Baduy community is driven by several factors, including general illiteracy, cultural practices, and time constraints. Consistent and long-term health programs are crucial to gradually overcoming these deep-rooted issues. Short-term interventions are unlikely to produce lasting change because they do not allow enough time to build trust, adapt to cultural nuances, or make significant improvements in literacy and health knowledge. Furthermore, the impacts of low health literacy, such as fatalism, high maternal and child mortality, and a vicious cycle of poor health maintenance, are severe and complex. These issues cannot be resolved with temporary solutions; they require ongoing education, support, and adaptation of health programs to the community’s evolving needs. Long-term partnerships can help build a resilient health infrastructure that continuously addresses these challenges.” |
|
4. Response to Comments on the Quality of English Language |
|
Point 1: The structure of sentences and paragraphs are often confusing, leaving space for multiple interpretations. English in itself is fine.
|
|
Response 1: We rewrite the Discussion section because we detected several confusing sentences and paragraphs in this section.
|
Reviewer 3 Report (Previous Reviewer 3)
Comments and Suggestions for Authors
The authors have incorporated most of the comments. The overall structure of the manuscript has improved, and fit for publication
One comment is that the limitation section (last page) should be shifted before the conclusion section.
Comments on the Quality of English Languageonly minor editing required
Author Response
Thank you for your positive feedback and for acknowledging the improvements made to the manuscript. We have taken note of your suggestion and move the limitations section to precede the conclusion. We appreciate your input in helping us refine the manuscript for publication
Round 2
Reviewer 2 Report (Previous Reviewer 2)
Comments and Suggestions for Authors
In this last version, the structure of the paper has improved substantially, and there are much fewer sentences open for multiple interpretations. Well done! I have no further main comments and think that this contribution from Indonesia is a very welcome addition to the existing literature.
I already informed the editor, but please have a look yourselves as well when the pdf arrived: check for small errors - a big disadvantage of the word format that MDPI uses is that editing of the text sometimes results in repeating or omitting words, in spaces being missing or excessively present, and that the citation numbers, when one reorders the text, are messed up. All of these happened in your text - which is not of scientific importance but it is worth it avoiding these little details. Surely sure the BMC typesetter can correct these mistakes to have a flawless text from the typesetting point of view and correct ordering of the references, but always better to double-check yourselves as well.
Comments on the Quality of English Languageno further comments just to check the repetition/omission of words, spaces etc after copying/pasting in the word document.
Author Response
Please see the attachment. Thank you.

This manuscript is a resubmission of an earlier submission. The following is a list of the peer review reports and author responses from that submission.
Round 1
Reviewer 1 Report
Comments and Suggestions for Authors
The manuscript entitled “Tackling Health Illiteracy of cultural-shocked indigenous population of Outer Baduy in Indonesia: the causes, effects, and alleviation” was interesting. The authors aimed to determine the causes, impacts, and solutions for health literacy in a village in Indonesia. The following comments can help them to improve the paper.
1- It is useful if the authors expand the introduction and review other materials related to health literacy in similar communities around the world.
2- The contents of the “Theoretical foundation” seems to be fragmented. Moreover, it can be merged into the introduction section to make it stronger.
3- As the aim of the study has been explained in the introduction section, it is not necessary to mention it again in the methods section.
4- If the researchers conducted a case study, it is better to mention it in the title (second part) and in the methods section clearly.
5- The number of the participants is limited. This point needs to be addressed in the limitation section.
6- Themes, subthemes and categories need to be presented in a table.
7- The discussion section needs to be expanded to compare the results with other similar articles.
8- Please add a limitation section to the manuscript.
Author Response
|
Comments 1: it is helpful if the authors expand the introduction and review other materials related to health literacy in similar communities around the world |
||||||||||||||||||||||||||||||||||||
|
Response 1: We expand the background by adding one paragraph discussing health literacy issues in other communities. The paragraph is: Research in several indigenous communities, such as the First Nation in Canada (Rheault et al., 2021) and Aboriginal and Torres Strait Islanders in Australia (Smith et al., 2020), shows that health literacy is an essential predictor of the quality of health of indigenous people. Unfortunately, indigenous people are vulnerable communities which are difficult to reach through modern health promotion and education efforts due to various cultural, language and policy barriers, as well as economic isolation and inequality (Arrighi et al., 2022)."
|
||||||||||||||||||||||||||||||||||||
|
Comments 2: The contents of the "theoretical foundation" seem fragmented. Moreover, it can be managed in the introduction section to make it stronger. |
||||||||||||||||||||||||||||||||||||
|
Response 2: Thank you. Previously, the theoretical foundation consisted of health literacy and the Baduy community. We cut the health literacy section and pasted it under the map before explaining the problem faced by the community. In this way, the section will make the issue stronger. Because the theoretical foundation now only consists of the Baduy community description, the section's title changed to Baduy community to highlight the context of this research.
|
||||||||||||||||||||||||||||||||||||
|
Comments 3: As the aim of the study has been explained in the introduction section, it is unnecessary to mention it again in the methods section. |
||||||||||||||||||||||||||||||||||||
|
Response 1: Agree, the section was erased.
|
||||||||||||||||||||||||||||||||||||
|
Comments 4: If the researchers conducted a case study. It is better to mention it in the title (second part) and the methods section. |
||||||||||||||||||||||||||||||||||||
|
Response 4: The title "Tackling Health Illiteracy of Cultural-Shocked Indigenous Population of Outer Baduy in Indonesia: the Causes, Effects, and Alleviation" has now changed to "Tackling Health Illiteracy of Cultural-Shocked Indigenous Population: A Case Study of Outer Baduy in Indonesia " The subsection "Design" in the methods section explains our choice of case study methodology.
|
||||||||||||||||||||||||||||||||||||
|
Comments 5: The number of participants is limited. This point needs to be addressed in the limitation section. |
||||||||||||||||||||||||||||||||||||
|
Response 5: we add this paragraph at the end of the conclusion: "The limited number of interviews and participants in this research indicates a constraint of the study. Although this is common in case study research, the small sample size prevents broader generalizations. Future studies should aim to include a larger number of participants to enhance the robustness of their findings."
|
||||||||||||||||||||||||||||||||||||
|
Comments 6: Themes, subthemes, and categories must be presented in a table. |
||||||||||||||||||||||||||||||||||||
|
Response 6: Table 1 was created for themes, subthemes, and categories.
|
||||||||||||||||||||||||||||||||||||
|
Comments 7: The discussion section needs to be expanded to compare the results with other similar articles |
||||||||||||||||||||||||||||||||||||
|
Response 7: Previously, we only mentioned two research studies and only stated that this research agrees with those previous studies. Now we add four more research studies and explain the similarities and differences. We say the similarity of the results of this research with a study in Nigeria (Oredola et al., 2020) in terms of imitation factor and with a study in Australia (Wynne et al., 2023) in terms of geographical factor. We contrast this research's findings with those from Australia (Nash and Arora, 2021) and South Africa (Ramulondi et al., 2021). Food taboos are not a significant factor in this research, which differs from this Research on Baduy.
|
||||||||||||||||||||||||||||||||||||
|
Comments 8: Please add a limitation section to the manuscript |
||||||||||||||||||||||||||||||||||||
|
Response 8: A limitation section was added at the end of the paper. Besides mentioning the limitation on the number of participants, we also acknowledge the language barrier and the use of a translator. |
Please see the attachment below. Thank you.

Reviewer 2 Report
Comments and Suggestions for Authors
Considering that this work studies effects of health literacy using stunting as a specific example, I suggest to mention stunting in the title.
Introduction: the second two-thirds of the introduction has no citations to the literature at all, while they should be there, please include them.
Methods: to be a case study, the combination of studying one youtube video and semi-structured interviews seems to be a bit meager, particularly as you list a mix of 6 methods often used in case studies.
Methods: you describe health illiteracy as not being able to read/write about health issues. However, many health illiteracey interventions do not aim at written materials but alternatives – as is also clear from the next sentences in your methods section. Health illiteracy, in my reading, is understanding things related to health (also non-written things, such as information received by health authorities through personal meetings, radio emission, infographics, etc.), and understanding how they affect your health or that or your close surroundings, allowing you to make choices that could positively affect your health. So I would suggest to revise this definition of health illiteracy in the second phrase of the methods section.
There is a paragraph that begins with “The study also revealed that the health issue that is easiest to communicate is the issue of death, followed by the issue of illness.” This seems to be results of a specific study but the reader has no idea where that study came from, the previous paragraphs are more general definitions and do not mention any study. Moreover, working myself in end of life issues, talking about death is repeatedly found to be incredibly difficult to talk about – do you mean here that the physician informs that a patient has died and that the information is clear or do you mean talking about death? Please clarify.
This section is confusing to me (There is a gender bias in health communication because, generally, women are the most comfortable people to discuss health topics with compared to men [14]. So, even though health information and communication that is considered the most reliable comes from the government [15], especially in areas that deal with health problems, the communication process occurs at the family level.): So the most reliable information comes from the government but the communication happens within families and women communicate better than men. Are women better at expressing their own health problems or are they better listing to and communicating about health problems of others? Is the health information from the government considered “dealing with health problems” or is that meant to be information to prevent health problems? There seems to be a mix of related topics knitted together too directly to follow the logic, please improve.
The information about the community is very important and useful, but there are some terms that are unnknown to many readers. For example, what is a “mandala”?
The objective of the study should probably state “This research aims to determine the causes, impacts, and solutions for health ILliteracy in the Baduy Luar village in Kanekes Village, Indonesia”, rather than to study causes of health literacy.
It is not clear from the methods if the 10 key persons were ten people and then snowballing occurred, or how many (and who) were the key people that snowballing began with and which were the criteria to select the first. Reorganising the paragraph to state first they key persons and the criteria for considering these as key perons, and how many were then identified through snowballing.
Also, it is written that “These critical sources include the head of the SRI Non-Governmental Organization and the village midwife.” This makes me doubt if these 2 were THE critical sources (and are they the same as the key sources) or were there more (as it states “include” it seems to imply that there were more critical persons than these two – if that is true, please describe all of them).
The following sentence seems to come from a protocol rather than from the paper presenting the results: as you have already analysed the data you know if you reached saturation and if there were ten or more sources. “The number of ten sources is still tentative because it depends on theoretical saturation. If saturation is reached with these ten people, where no more new information can be abstracted even though the number of sources increases, data collection is stopped.”
Related to this, please also describe how you are going to evaluate saturation: usually mentioning an X number of observations without adding new information, but that implies analyses on the go etc – such information is incomplete.
“The five resource persons knew the researcher before data collection because they provided the initial data needed to understand the problems faced by the Baduy community” – which 5 resource persons? Were these the key persons? So then there were 3 more than the two mentioned? Please reorganise the part on the participant invitation, selection and acceptance so that the reader can understand. It would be useful to make a diagram showing who were initially invited, who came through snowballing, who were already know to the researchers and who did not know them, and the categories in which they are (community members, community leaders and external stakeholders).
The text in this section is written in present tense but should be past tense in many phrases.
Please describe a bit more on the youtube video, who made it, why, which was the objective, was the population involved in the design of it or not, etc.
Again, citations to the relevant methodological literature is lacking.
In ethical considerations – was any approval by ethics committee or similar authorities obtained?
Results:
There are very interesting topics in the results, which do not go deeper than the phrase, for example on the gender issue – where it is stated that women are not considered to know MORE than men but the citation is that it seems that they feel like it is the men who should know more (so not aim for equal or even better knowledge amongst women but probably that the men should know).
Also, the fact of the prohibited food items is probably a clue for adapted health messages – not recommended to eat chicken but other nutritious foods that could meet with the dietary requirements and avoid stunting rather than the standard message by the government – I presume that in the interviews it was asked if the communicators received an adapted message for advise for this community of if they feel that this could be useful.
Regarding the sad observations regarding fatalism – presumably they were asked if they feel that under other circumstances their health as a community could be better, if the people could grow taller, have less child mortality? What were the answers to such questions?
Several citations leave many questionmarks – for example this one (but there are more): Violations of various kinds are reported, and then there are instructions, such as if you eat food, you must be careful – which food should you be careful with and what is “being careful”? Is it related to not eating prohibited items or on preparing/cooking it well to avoid infections, is it something else?
The part on the breast cancer is very confusing: is this information that is being passed on correct? And I have no idea what this sentence means “Then, in some mammary cases, breast cancer appears, and many of them die and are not exposed. Because of that, their lifestyle changed."” Who are the “mammary cases”, who die and are not exposed to what? Lifestyle of who changed? Of the patients, of the population? Towards which kind of consumption?
Which observations came from the interviews, which from the youtube video? You describe in the methods that you triangulate the results: where did the observations coincide, differ, how were these interpreted/resolved?
Discussion: please do not mention that these factors “cause” low literacy – they are related and may be associated with, causally causing or being caused by low health literacey”. Many of the factors listed are probably related to difficulties in improving the health literacy but not causing it directly.
The prohibition of foods combined with low income makes it difficult for the Baduy people to be able to obtain nutritious diets – so then maybe they have the literacy but not the means to change their diets – their beliefs/religion does not allow them the cheaper proteins and the costs of the alternatives does not allow them the alternatives. Then the intervention is not on health literacy but on providing cheap nutritious alternatives that are healthy and allowed by their belief system. Partially this could also apply to the fatalism: in other parts of the world there are people who know about protecting health but they do not have the means to have the access, so that still leaves them in the fatalism, as they cannot improve their situation. These kinds of very complex mechanisms in reasoning and observing the situation could be deepend in the discussion section.
In the discussion there is no mention of the role of the interviewer, on strenghts and limitations, nor on how and if saturation was reached (and on which topics). The results can and should also be placed in the broader international literature on health illiteracy, stunting problems and addressing these in certain ethnical minority groups with different beliefs and world views.
The conclusion includes a word of thanks… which is not the appropriate place.
Minor comments.
Please check your text for incomplete sentences, such as for example this one, in which it is not clear who are these stakeholders or what happens to them: ¨Stakeholders with the Baduy community, which is a community that experienced a shock in their cultural contact with modern society.:
In many places in the text it seems that the authors are giving a class about the methodology - not all the reasons and details of the methods need to be explained, but rather HOW these concepts were measured/evaluated (and the results of them - now the methods have a whole range of quality evaluations mentioned that are not mentioned in the results, such as dependability, confirmability, transferability, and credibility issues - but in the results their evaluation is not mentioned at all)
Comments on the Quality of English LanguageThe English in itself is quite alright, but the organisation of the text into a coherent, logical structure with a flow that is easy to follow for the reader should be improved.1
Author Response
Please check the attachment below. Thank you.

Reviewer 3 Report
Comments and Suggestions for Authors
This research addresses a critical issue of health illiteracy among indigenous population. The research aims to determine the causes, impacts, and solutions for health literacy in one of the indigenous communities in Indonesia. The research has been conceived and executed and presented in scholarly manner. The findings contribute to the existing knowledge. However, the following comments are to be considered by the authors:
Introduction
1. Line 53-54 - There is some disconnect in this sentence. This sentence is abruptly used which affects the flow. “Appropriate health communication strategies are essential in this research because the above conditions persist today, even though many parties are trying to improve the 54 problem of stunting and general health in the Baduy community”
2. Line 55 -66 - Please briefly mention the major stakeholders here.
3. Line 62-63 the sentence “Most programs carried out by stakeholders are hit and run because they are only done once or twice, not prolonged, so they do not have long-term effects”- this sentence is not clear. Please give some examples of these programs.
4. In introduction section, please provide some details about the Baduy community – their population and other socio-economic characteristics.
Theoretical Foundation
1. Line 111 -112 the sentence “The study also revealed that the health issue that is easiest to communicate is the issue of death, followed by the issue of illness” not clear. Which study does it reveal?
2. Please specify the theory, on which this research is based?
Methodology
1. Data Collection - Brief details of information collected through semi-structured interview schedule
2. Line 233 – “The Baduy community consists of two mothers who have stunted children (Baduy 1 and Baduy 2)” - sentence is not clear. Is it mean only two mother have stunted children in whole community? .If s out of how many mothers?
3. How was the secondary data used?
Results
1. It is suggested that authors may use tables to show the causes and impacts, based on the findings.
Discussion
1. In discussion section, it is also suggested that authors may corroborate the finding of this study with more studies on this aspect, if possible.
2. Any limitations / challenges faced in conducting this research? or any other limitation encountered in this work. If so please give details.
Conclusion
1. Policy implications from the government side (on alleviating health illiteracy based on study findings) and need for any future research if any ? should be highlighted.
Author Response
Please see the attachment, thank you.
